# Strategy and Mechanism of Rice Bran Protein Emulsion Stability Based on Rancidity-Induced Protein Oxidation: An Ultrasonic Case Study

**DOI:** 10.3390/foods11233896

**Published:** 2022-12-02

**Authors:** Qi Zhou, Helin Li, Fang Li, Benpeng Zhang, Xiaojuan Wu, Wei Wu

**Affiliations:** 1College of Food Science and Engineering, Central South University of Forestry and Technology, Changsha 410004, China; 2National Engineering Research Center of Rice and Byproduct Deep Processing, Changsha 410004, China

**Keywords:** emulsion stability, interface protein, protein oxidation, ultrasonic treatment

## Abstract

To provide a strategy for improving the stability of rice bran protein emulsion (RBPE), rice bran proteins (RBPs) with different oxidation extents were prepared from fresh rice bran (RB) stored for different times (0, 1, 3, 5, 10 d), and RBPE was prepared with ultrasonic treatment. The ultrasonic conditions were optimized according to the results of the RBPE’s stability (when RB stored for 0, 1, 3, 5, 10 d, the optimal ultrasonic treatment conditions of RBPE were 500 w and 50 min, 400 w and 30 min, 400 w and 30 min, 300 w and 20 min, 500 w and 50 min, respectively). Additionally, the structural characteristics and the flexibility of RBPE interface protein were characterized, and the results showed that compared with native protein and excessive oxidized protein, the unfolded structure content and flexibility of interface protein of RBPE prepared by moderate oxidized protein under optimal ultrasonic intensity was higher. Furthermore, the correlation analysis showed that the RBPE stability was significantly correlated with the structural characteristics and flexibility of the RBPE interface protein (*p* < 0.05). In summary, ultrasonic treatment affected the interface protein’s structural characteristics and flexibility, improving the stability of RBPE prepared from oxidized RBP.

## 1. Introduction

Rice (*Oryza sativa* L.) is a staple food for more than half of the world’s population, and rice bran (RB) is a major byproduct of rice processing with an annual global production of 29.3 million tons per year [1]. The main components of RB include 14–16% protein, 12–23% lipid, and 8–10% dietary fiber [2]. Rice bran protein (RBP) is a high-quality protein with a balanced amino acid composition [3]. RBP exhibits anticancer activity, hypoallergenicity, and higher digestibility [4]. In addition to nutritional value, RBP also has excellent emulsifying properties [5]. However, RB is prone to rancidity during rice processing and storage, resulting in a harsh taste and unfavorable flavor, limiting the application of RBP in the food industry [6,7].

The RB contains highly active endogenous lipase; RB rancidity inevitably occurs during rice milling, which further leads to protein oxidation [6]. Protein oxidation induced by RB rancidity can change protein structure, thus affecting protein emulsion stability [6]. The moderate oxidation of RBP can cleave the disulfide bonds, enhance the flexibility of RBP structure, and promote the unfolding of RBP structure [5,8]. Structural modifications hastened the spread and adsorption of RBP at the oil–aqueous interface and increased the viscoelasticity of the interface membrane, improving the stability of the rice bran protein emulsion (RBPE) [5]. The excessive oxidation of RBP can result in the formation of disulfide bonds, a decrease in flexibility of RBP, and promotion of aggregation [9]. Structural changes can inhibit the spread and adsorption of RBP at the interface, and decrease the viscoelasticity of the interface membrane, resulting in RBPE destabilization [5]. Therefore, the oxidation extent of protein is a nonnegligible factor in the optimization of emulsion stabilization strategies in the production of protein emulsions.

The industrial production of protein emulsion is usually obtained by high-pressure homogenization, microfluidization, and ultrasonic treatment [10,11]. Ultrasonic treatment is not only energy-efficient and simply manipulated, but also has a significant impact on protein structure and flexibility [12]. Optimal ultrasonic intensity can expose active sites without affecting structural integrity of the protein, thus enhancing protein flexibility and improving emulsifying properties [13,14]. Theoretically, ultrasonic treatment, as an emulsion stabilization strategy, has great potential to improve the structural and emulsifying properties of oxidized RBP induced by RB rancidity. However, few studies have been conducted on the effects of ultrasonic treatment on the emulsifying properties of oxidized protein (specifically oxidized RBP induced by RB rancidity). Therefore, the ultrasonic intensities (the combination of ultrasonic power and treatment time) to prepare RBPE from RBP with different oxidation extents were optimized according to the emulsion stability results. The interface protein structure and flexibility of RBPE prepared from RBP with different oxidation extents were analyzed. This research aims to provide a strategy for improving the stability of RBPE prepared from RBP with different oxidation extents and to investigate the related potential mechanism by exploring the correlation between changes in structural characteristics/flexibility of the interface protein and the emulsion’s stability.

## 2. Materials and Methods

### 2.1. Materials

Fresh RB was bought from Ningxiang County Jinquan Rice Factory, the variety of corresponding rice is Huang Hua Zhan (Changsha, Hunan, China). Electrophoresis protein standards were provided by TransGen Biotech (Beijing, China), and other reagents were provided by Sinopharm Chemical Reagent Co., Ltd. (Beijing, China).

### 2.2. RBP Extraction

Fresh RB was stored for 0, 1, 3, 5, and 10 d to prepare the RB with different rancidity extents (25 °C, relative humidity 85%). The RB was soaked in petroleum ether at 1:5 (*w/v*). Shaked the mixture for 30 min before suction filtration, then obtained the filter cake (defatted RB) and the filtrate. Petroleum ether was recovered, and crude RB oil was collected using a rotary evaporator. The defatted RB flours were stored in the refrigerator (4 °C). Defatted RB and deionized water were mixed at 1:10 (*w*/*v*), and then the pH was adjusted to 9.0. After stirring the solution for 4 h, the mixed solution was centrifuged. The pH of the supernatant was adjusted to 4.0, and left to stand for 35 min, then centrifuged to obtain the RBP precipitate. The RBP precipitate was dispersed with deionized water, the pH adjusted to 7.0, and then dialyzed with deionized water for 24 h. Finally, the RBP was obtained by freeze drying. The protein content of RBP prepared by RB stored for 0, 1, 3, 5, 10 d was 89.77 g/100 g, 89.63 g/100 g, 90.26 g/100 g, 89.67 g/100 g, 89.41 g/100 g (dry basis), measured based on the micro-Kjeldahl method.

### 2.3. Measurement of RB Rancidity Extent and RBP Oxidation Extent

Determination of the acid value of RB crude oil was carried out, referencing the standard method of AOCS Cd 8–53. The measurement of carbonyl content was based on the method of Li et al. [5]. 2,4-dinitrophenylhydrazine was blended with 0.35 mL RBP solution to measure the carbonyl group. The molar extinction coefficient was 22,000 (mol/L)^−1^ cm^−1^.

### 2.4. Preparation of RBPE

According to the method of Zhang et al. [15], the RBP was dissolved in phosphate buffer (0.01 mol/L, pH 7.0) to obtain a 10 mg/mL RBP solution. Soybean oil was added to the RBP solution at a ratio of 1: 4 (*v*/*v*), then went via high-speed shearing (T18basic, IKA, UK) at 15,000 rpm for 2 min, then was treated with the ultrasonic homogenizer (SCIETZ-IID, Ningbo Scientz Biotechnology Co., Ltd., Ningbo, China) in an ice bath at 300 W, 400 W, and 500 W with a Φ10 mm ultrasonic probe for 20 min, 30 min, 40 min, and 50 min, respectively. The volume of the mixtures for a single treatment was 20 mL. The sample without ultrasonic treatment could not form a stable emulsion and layered after standing for a period of time; therefore, the sample without ultrasonic treatment was not included in the present research.

### 2.5. Determination of Emulsion Stability

#### 2.5.1. Mean Droplet Diameter, Polydispersity Index, and Zeta Potential Determination of RBPE

The mean droplet diameter (MDD) and polydispersity index (PDI) were diluted with deionized water (0.1%, *v/v*), and the zeta potential was diluted with phosphate buffer (0.2%, *v/v*) and then determined via the particle size analyzer (Nano ZS, Malvern Instruments Ltd., Worcestershire, UK), which was based on the procedure of Chen et al. [16]. 

#### 2.5.2. Microstructure Measurement of RBPE

The RBPE was diluted 10-fold with phosphate buffer, then evaluated using a biological microscope. RBPE (10 μL) was placed on a glass slide with coverslip. The objective magnification was a 40 lens.

#### 2.5.3. Creaming Index Measurement of RBPE

The determination of the creaming index (CI) was referenced in the procedure of Li et al. [5]. RBPE (8 mL) was added into the tube (20 mm diameter × 50 mm height) and stored for 7 d. The calculated equation was:CI (%) = H_1_/H_0_ × 100(1)
where H_1_ was the height of the aqueous phase, and H_0_ was the height of RBPE.

### 2.6. Extraction of Interface Protein 

The method of extracting interface adsorbed protein (IAP) and interface non-absorbed protein (INP) was based on the description by Li et al. [5]. The RBPE was centrifuged and washed to obtain the IAP precipitate. The phosphate buffer for washing the precipitate was collected for freeze-drying to obtain the INP, and the precipitate was mixed with cold acetone (−40 °C). After standing for 2 h, the mixture was centrifuged and washed 3 times to collect the precipitate. Finally, the precipitate was dispersed in water for freeze-drying to obtain the IAP.

### 2.7. IAP Content Measurement

IAP content (Γ) was measured based on the procedure of Yi et al. [17]. INP concentration (aqueous phase) was calculated using Bradford’s method. The equation was:Γ (mg/m^2^) = [(1 − φ) d_3,2_/6φ] (C_1_ − C_2_)(2)
where C_1_ and C_2_ were the initial RBP concentration and residual RBP concentration of the aqueous phase, and φ was the oil phase volume fraction.

### 2.8. Methods for Determining the Structure of IAP and INP

#### 2.8.1. Sulfhydryl and Disulfide Content Measurement

Contents of free sulfhydryl and disulfide bonds were measured using the 5, 5’-Dithiobis (2-nitrobenzoic acid) colorimetric method. RBP concentration was determined using Bradford’s method, and RBP content was determined using the Kjeldahl method. The extinction coefficient was 13,600 mol^−1^ cm^−1^. 

#### 2.8.2. Fourier Transform Infrared Spectroscopy (FTIR) Measurement

The FTIR method was based on the procedure of Li et al. [5]. The protein and KBr were mixed at 1:100 (*w*/*w*). Then, the mixture was ground to produce transparent flakes. The flakes were placed in an IRTracer-100 spectrometer (Shimadzu, Kyoto, Japan) to determine the FTIR. The spectra were scanned from 1600 to 1700 cm^−1^ with 128 scans.

#### 2.8.3. Flexibility Measurement

The flexibility of IAP and INP was determined according to a method described by Li et al. [5]. The centrifugation parameters were 4000× *g* and 30 min, and the absorbance was 280 nm.

#### 2.8.4. Size Distribution and Zeta Potential Measurement

Size distribution and zeta potential of IAP and INP were measured according to the method described by Chen et al. [18]. The IAP and INP were blended in phosphate buffer to obtain the IAP and INP with a concentration of 1.0 mg/mL, respectively.

#### 2.8.5. Surface Hydrophobicity Measurement

The method of Li et al. [5] was used to determine the surface hydrophobicity of IAP and INP. The instrument used for the measurement was an F-7000 fluorescence spectrometer (Shimadzu, Kyoto, Japan). The excitation and the emission wavelengths were 390 and 470 nm, respectively.

#### 2.8.6. Intrinsic Fluorescence Measurement

Based on the description of Li et al. [5], the protein was dissolved in a phosphate buffer with a concentration of 0.1 mg/mL. The intrinsic fluorescence of IAP and INP was determined by the F-7000 fluorescence spectrometer (Shimadzu, Kyoto, Japan). The excitation and emission wavelengths were 295 nm and 300–500 nm, and the scanning speed was 10 nm/s.

#### 2.8.7. Protein Electrophoresis 

Following the method of Wu et al. [6], the concentrations of the stacking gel and the separating gel were 40 and 125 g/L, respectively. The electrode buffer contained 0.1% sodium dodecyl sulfate, 0.384 mol/L Gly, and 0.05 mol/L Tris. The sample solution contained 0.01 mol/L Tris-HCl buffer (pH 8.0), 2% sodium dodecyl sulfate, 10% glycerol, 0.02% bromophenol blue, and 5% β-mercaptoethanol. The loading volume was 10 μL. Electrophoresis was 10 mA at the beginning and 25 mA after the sample entered the separating gel.

### 2.9. Statistical Analysis

All data were analyzed with the SSPS statistical analysis program and significant differences were evaluated by Duncan’s multiple range tests in SPSS (*p* < 0.05).

## 3. Results and Discussion

### 3.1. Effect of RB Storage Time on the RB Rancidity Extent and RBP Oxidation Extent

As shown in Figure 1, with the prolongation of RB storage time, the acid value of RB crude oil and the carbonyl content of RBP increased significantly (*p* < 0.05). The phenomena indicated that during RB storage, RB rancidity and RBP oxidation simultaneously occurred. RB contained highly active lipase, which could rapidly catalyze the hydrolysis of lipids to form free fatty acids. The free fatty acids were unstable and easily oxidized to produce various reactive oxygen species, which resulted in protein oxidation [5].

### 3.2. Effect of Ultrasonic Treatment on the Stability of RBPE Prepared from RBP with Different Oxidation Extents 

#### 3.2.1. Analysis of MDD, Size Distribution, and Zeta Potential 

MDD was a key factor in evaluating emulsion stability; PDI and zeta potential were usually associated with the homogeneity of particle size distribution and electrostatic repulsion, respectively [19]. Results for the MDD, PDI, and zeta potential of RBPE were used to optimize ultrasonic conditions for preparing RBPE. As shown in Table 1, when RB storage time was 0 d, the optimal ultrasonic condition was power: 500 W and time: 50 min. When RB storage time was 1 d and 3 d, the optimal ultrasonic condition was power: 400 W and time: 30 min. When RB storage time was 5 d, the optimal ultrasonic condition was power: 300 W and time: 20 min. However, when RB storage time was 10 d, the optimal ultrasonic condition was power: 500 W and time: 50 min. The phenomena indicated that with the increase of oxidation extent, the ultrasonic intensity required for the optimal RBPE initially decreased and then increased.

Previous research in our laboratory found that moderately oxidized RBP induced by RB rancidity could enhance the flexibility of RBP, unfold the structure of RBP, and improve the rate of protein spread to the oil–aqueous interface, thereby improving the stability of the RBPE [5]. The energy provided by the ultrasonic treatment can unfold the protein structure, and excessive energy induces protein aggregation [20,21]. Therefore, the optimal condition of RBPE prepared by moderately oxidized RBP (extracted from RB stored for 5 d) is mild ultrasonic intensity, and excessive ultrasonic intensity shows negative effects on the stability of RBPE. Compared with the moderately oxidized RBP, the excessively oxidized RBP (extracted from RB stored for 10 d) exhibits increased aggregation and elevated rigidity [5]; thus, strong ultrasonic intensity was required to unfold the RBP structure and enhance RBP flexibility

#### 3.2.2. Analysis of Macroscopic Stability and Microstructure

The CI of the emulsion was related to the ability of the emulsion to resist the changes of external environment, and the microstructure also intuitively reflected the distribution of emulsion droplets. Results of CI and microstructure of RBPE were used to optimize ultrasonic conditions preparing RBPE. As shown in Figure 2, when RB storage time was 0 d and 10 d, the optimal ultrasonic condition was power: 500 W and time: 50 min. When RB storage time was 1 d and 3 d, the optimal ultrasonic condition was power: 400 W and time: 30 min. When RB storage time was 5 d, the optimal ultrasonic condition was power: 300 W and time: 20 min. The optimized results were consistent with those of the MDD and zeta potential, indicating that compared with excessively oxidized RBP, the optimal condition for RBPE prepared by moderately oxidized RBP was mild.

When the protein is adsorbed on the oil–aqueous interface, steric hindrance and electrostatic repulsion inhibit the aggregation of droplets [5]. Owing to the lower unfolding extent of the native protein structure (RBP extracted from RB stored for 0 d), a strong ultrasonic intensity is required to unfold the protein structure and promote the steric hindrance and electrostatic repulsion between droplets [22]. Moderate protein oxidation can unfold the structure of the protein, which enhances the steric hindrance and electrostatic repulsion between droplets; therefore, the ultrasonic intensity required for preparing the optimal RBPE is mild [23]. Excessive protein oxidation might induce protein aggregation, thus reducing the steric hindrance and electrostatic repulsion between droplets [18]. Therefore, the preparation of optimal RBPE required strong ultrasonic intensity

### 3.3. Effect of Ultrasonic Treatment on the Content and Structure of IAP and INP of RBPE Prepared from RBP with Different Oxidation Extents 

According to the MDD, PDI, zeta potential, CI, and microstructure results, the optimal ultrasonic conditions for preparing RBP emulsion with different oxidation extents were obtained. When the storage time of RB for preparing RBP was 0, 1, 3, 5, and 10 d, the optimal treatment conditions for ultrasonic preparation of RBPE were 500 W and 50 min, 400 W and 30 min, 400 W and 30 min, 300 W and 20 min, 500 W and 50 min, respectively. The RBPE prepared under all optimal ultrasonic conditions was classified as group A. Correspondingly, the worst ultrasonic conditions for preparing the emulsion of RBP with different oxidation extents were also obtained. When the storage time of RB for preparing RBP was 0 d, 1 d, 3 d, 5 d and 10 d, the worst treatment conditions for ultrasonic preparation of RBPE were 300 W and 20 min, 300 W and 20 min, 300 W and 20 min, 500 W and 50 min, 300 W and 20 min, respectively. The RBPEs prepared under the worst ultrasonic conditions were classified as group B.

#### 3.3.1. Analysis of the IAP Content of RBPE Prepared from RBP with Different Oxidation Extents 

As shown in Table 2, the content of IAP in group A was significantly lower than that in group B (*p* < 0.05). The phenomena indicated that the optimal ultrasonic intensity led to the unfolding of the IAP structure in group A. In group B, excessive ultrasonic intensity led to the formation of protein aggregates at the oil–aqueous interface, while mild ultrasonic intensity failed to provide enough energy to unfold the structure of IAP, so the IAP content in group B was higher than that in group A [21]. The lower the content of IAP, the higher extent of protein unfolding, which could enhance the steric hindrance, inhibit the movement of droplets, improve the stability of RBPE, and thus obtain a smaller MDD and zeta potential [5]. The decrease in emulsion stability in group B might be due to IAP experiencing greater extrusion of Coulomb force, so IAP tended to be more rigid [20]. Li et al. [5] found that with the prolongation of RB storage time, the IAP content of RBPE first decreased and then increased, due to protein oxidation induced by rancidity. In this study, as the RB storage time prolonged, the IAP content of each group initially decreased and then increased. The phenomena indicated that although the optimal ultrasonic intensity could improve the rate of adsorption and arrangement of protein at the oil–aqueous interface by unfolding the protein structure and enhancing the flexibility of protein, it could not completely eliminate the negative effects of protein aggregation caused by excessive oxidation on emulsion stability.

#### 3.3.2. Analysis of the Content of Free Sulfhydryl and Disulfide Bonds of IAP and INP of RBPE Prepared from RBP with Different Oxidation Extents 

As shown in Table 2, the free sulfhydryl content of group A presented significantly higher than that of group B (*p* < 0.05), and the disulfide bond content showed an opposite trend, indicating the unfolded structure content of group A was higher. This might be due to the fact that the optimal ultrasonic intensity can break disulfide bonds, thus exposing more sulfhydryl groups [14], while in group B, the mild ultrasonic intensity could not effectively unfold the protein structure, and the excessive ultrasonic intensity induced protein aggregation, thus promoting the formation of disulfide bonds [22,24]. In addition, the content of free sulfhydryl in IAP was higher than that in INP in the two groups, indicating that IAP was more flexible than INP. It could be obviously observed that with the prolongation of RB storage time, the free sulfhydryl content of IAP in the two groups initially increased and then decreased, while the content of the disulfide bonds showed the opposite trend. This phenomenon might be due to protein aggregation caused by excessive oxidation, which could not be effectively unfolded under ultrasonic treatment. In the INP of the two groups, it can be observed that with the prolongation of RB storage time, the content of the free sulfhydryl group gradually decreased, and the content of the disulfide bond gradually increased, which might be due to the conversion of the free sulfhydryl group to other sulfur compounds [5].

#### 3.3.3. Analysis of FTIR of IAP and INP of RBPE Prepared from RBP with Different Extents of Oxidation

Figure 3A shows the percentage of secondary structures of IAP and INP in RBPE analyzed by FTIR. The α-helix represents the rigid secondary structure, the β-sheet represents the flexible secondary structure, and the ratio of α-helix/β-sheet reflects the flexibility of the protein secondary structure [5]. The α-helix content of IAP and INP in group A was significantly lower than that of group B, but the β-sheet content of IAP and INP in group A was significantly higher than that of group B (*p* < 0.05). This phenomenon indicated that flexibility of the secondary structure of IAP and INP in group A was higher than that of IAP and INP in group B, possibly because the optimal ultrasonic intensity could induce the unfolded secondary structure of the protein, breaking intramolecular hydrogen bonds and increasing intermolecular hydrogen bonds [14]. Excessive ultrasonic intensity caused protein aggregation, while mild ultrasonic intensity could not effectively unfold the protein, which was contrary to the optimal ultrasonic intensity [25]. There was no significant difference among the α-helix content of IAP and INP in the two groups, but the content of the β-sheet of IAP in the two groups was significantly higher than that of INP (*p* < 0.05). The phenomenon indicated that the flexibility of IAP was significantly higher than that of INP (*p* < 0.05), which might be due to the rapid formation of a viscoelastic membrane at the oil–aqueous interface of the flexible protein during the preparation of RBPE [5].

#### 3.3.4. Analysis of Surface Hydrophobicity, Zeta Potential, and Protease Susceptibility of IAP and INP of RBPE Prepared from RBP with Different Oxidation Extents 

Figure 3B shows the surface hydrophobicity of interface protein under different ultrasonic intensities. The surface hydrophobicity of IAP and INP in group A was significantly higher than in group B (*p* < 0.05), which might be due to the unfolding of protein spatial structure and exposure of hydrophobic amino acid residues by optimal ultrasonic intensity [7,26], while in group B, the excessive ultrasonic intensity led to protein aggregation, therefore burying the hydrophobic amino acid residues, and mild ultrasonic intensity could not effectively unfold the protein structure and release the hydrophobic amino acid residues [27,28]. The surface hydrophobicity of IAP was higher than that of INP in the two groups, which might be due to unfolded protein being more prone to adsorb to the oil–aqueous interface and form a viscoelastic membrane [5]. In addition, with the prolongation of RB storage time, the surface hydrophobicity of IAP and INP first increased and then decreased in the two groups. This phenomenon was similar to the results of the preparation of RBPE with different oxidation extents of RBP (range from 640.83 to 349.61) by high-pressure homogenization, which might be because moderate oxidation unfolded the RBP structure and exposed hydrophobic amino acid residues, while excessive oxidation led to protein aggregation and buried the originally exposed hydrophobicity amino acid residues [5]. Moreover, the surface hydrophobicity of the RBPE with different oxidation extents prepared by ultrasonic treatment was significantly higher than that prepared by high-pressure homogenization, indicating that ultrasonic treatment improved the surface hydrophobicity of RBP, and the improvement effect on moderately oxidized RBP was the most significant.

As shown in Figure 3C, the zeta potential of IAP and INP in group A was significantly lower than that in group B (*p* < 0.05), which might be ascribed to the optimal ultrasonic intensity-unfolded protein and exposed charged amino acids to the protein surface, thus enhancing the electrostatic repulsion between proteins [29]. The zeta potential of IAP in both groups was lower than that of INP, which indicated that the IAP structure was more unfolded than INP. This phenomenon might be due to the fact that highly unfolded protein is more prone to adsorbing to the oil–aqueous interface [5]. Moreover, with the prolongation of RB storage time, the zeta potential of IAP and INP in both groups initially decreased and then increased, which might be ascribed to the inability of protein aggregates caused by excessive oxidation to effectively unfold under optimal ultrasonic intensity.

As shown in Figure 3D, the protein flexibility in group A was significantly higher than that in group B (*p* < 0.05), which might be because the optimal ultrasonic intensity-unfolded protein structure, thus exposing more active sites and increasing flexibility [8,30]. However, excessive ultrasonic intensity led to protein aggregation, thereby masking the originally exposed active sites, while the mild ultrasonic intensity could not effectively unfold the protein aggregates [8,30]. In addition, the flexibility of IAP was higher than that of INP, which might be due to the unfolded protein absorption on the oil–aqueous interface and the process of unfolding and arranging under ultrasonic treatment, leading to the formation of a viscoelastic membrane [5,31].

#### 3.3.5. Effect of Ultrasonic Treatment on the Spatial Structure of IAP and INP of RBPE Prepared from RBP with Different Oxidation Extents

##### Analysis of the Size Distribution and Intrinsic Fluorescence of IAP and INP of RBPE Prepared from RBP with Different Oxidation Extents

The size distribution is shown in Figure 4A,B and the intrinsic fluorescence spectra of interface protein are shown in Figure 4C,D. The wavelength of maximum emission of tryptophan fluorescence (λ_max_) was sensitive to its local environment. The MDD of IAP and INP in group A was significantly lower than that in group B (*p* < 0.05), and the λ_max_ of IAP and INP in group A was larger than that in group B, indicating that IAP and INP aggregated in group B. This phenomenon might be attributed to the optimal ultrasonic intensity-unfolded protein structure, but the excessive ultrasonic intensity led to protein aggregates, while mild ultrasonic intensity could not effectively unfold the protein structure [20,24]. In addition, the MDD of INP in the two groups was greater than that of IAP, and the λ_max_ had an opposite trend, which might be due to the fact that highly unfolded protein is prone to arranging on the oil–aqueous interface during RBPE formation, forming a viscoelastic membrane [5]. The MDD of IAP and INP in both groups initially decreased and then increased with the RB storage time prolonged. This phenomenon indicated that the structure of the moderately oxidized protein could be unfolded at optimal ultrasonic intensity, but the aggregates of the excessively oxidized protein could not effectively unfold at optimal ultrasonic intensity. In addition, with the prolongation of RB storage time, the intrinsic fluorescence intensity of each group gradually decreased, which might be related to the oxidation of tryptophan induced by RB rancidity [5].

##### Analysis of the SDS-PAGE of IAP and INP of RBPE Prepared from RBP with Different Oxidation Extents

Figure 4E–H shows the electropherograms of IAP and INP in the two groups, respectively. The molecular weight of the interface protein subunits of RBPE mainly ranged from 7 to 80 kDa [5,32]. There was no change in the number of bands in reducing and non-reducing electrophoresis, indicating that the subunits of IAP and INP did not change during ultrasonic treatment and RB storage [6]. Compared with the reduced electropherogram, the aggregation bands and the prolamin subunit bands became lighter. The phenomenon showed that the disulfide bonds were cleaved by β-mercaptoethanol in the reduced electropherogram [6]. The aggregation bands, albumin subunit bands, and glutelin acidic subunit bands of IAP and INP in group B were darker than those in group A, suggesting that the protein structure in group B was more aggregated [5].

### 3.4. Correlation Analysis

In order to determine the relationship between RBPE stability and the structural characteristics of IAP and INP, the correlation analysis between MDD, PDI, zeta potential, and CI of RBPE and the disulfide bonds, α-helix/β-sheet ratio, flexibility, surface hydrophobicity, particle size distribution, and intrinsic fluorescence of IAP and INP was analyzed (Table 3). The MDD, PDI, zeta potential, and CI of RBPE were significantly positively correlated with zeta potential, and particle size distribution of IAP and INP (*p* < 0.01), which indicated that the higher extent of the unfolded protein structure was, the better the stability of RBPE was. The MDD, PDI, zeta potential, and CI of RBPE were negatively correlated with the α-helix/β-sheet ratio, disulfide bonds’ content flexibility, and surface hydrophobicity of IAP and INP (*p* < 0.01), which indicated that the flexibility of interface protein presented a positive correlation with the stability of RBPE. There was no significant (*p* > 0.05) correlation between the MDD, PDI, zeta potential, and CI of RBPE and intrinsic fluorescence of IAP and INP. The results of the correlation analysis show that the structure and flexibility of RBPE interface protein were important factors affecting the stability of the emulsion.

## 4. Conclusions

The effect of RB rancidity-induced RBP oxidation on RBPE stability was improved by adjusting the strategy for preparing stable RBPE. Ultrasonic treatment provided targeted improvements to the stability of RBPE with different extents of oxidation by unfolding the interface protein structure and increasing the interface protein flexibility. Interestingly, compared with native RBP and the excessively oxidized RBP, the structure of the moderately oxidized RBP was more unfolded and flexible, which increased steric hindrance and electrostatic repulsion between the emulsion droplets; thereby, the ultrasonic intensity for preparing corresponding optimal RBPE was mild. The stability of RBPE was significantly correlated with the structural characteristics and flexibility of the RBPE interface protein. The results of the present study provide a strategy for improving the stability of RBPE prepared from oxidized RBP, and also provide theoretical reference and technical guidance for the preparation of emulsion from oxidized protein by ultrasonic treatment.

## Figures and Tables

**Figure 1 foods-11-03896-f001:**
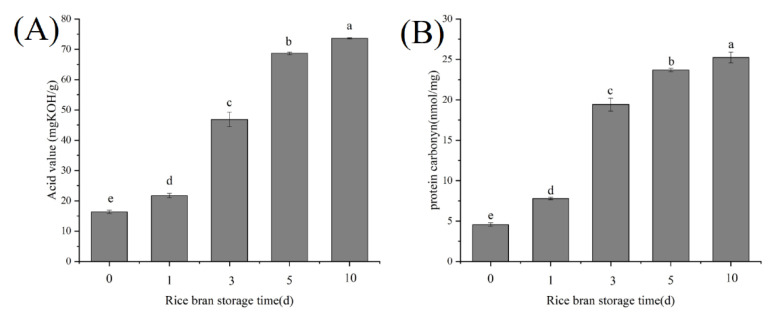
Effects of RB storage time on the acid value of RB crude oil (**A**) and RBP carbonyl content (**B**), different lowercase letters within a column indicate significant differences (*p* < 0.05).

**Figure 2 foods-11-03896-f002:**
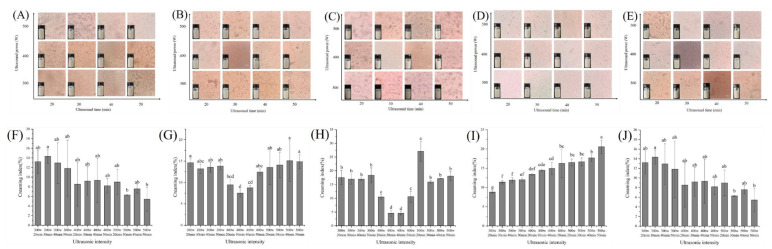
Effects of ultrasonic intensity on microstructure for stored 0 d of RBPE prepared from RB with different storage times ((**A**)—0 d, (**B**)—1 d, (**C**)—3 d, (**D**)—5 d, (**E**)—10 d) and effects of ultrasonic intensity on CI for stored 7 d of RBPE prepared from RB with different storage times ((**F**)—0 d, (**G**)—1 d, (**H**)—3 d, (**I**)—5 d, (**J**)—10 d). The different lowercase letters within a column indicate significant differences (*p* < 0.05).

**Figure 3 foods-11-03896-f003:**
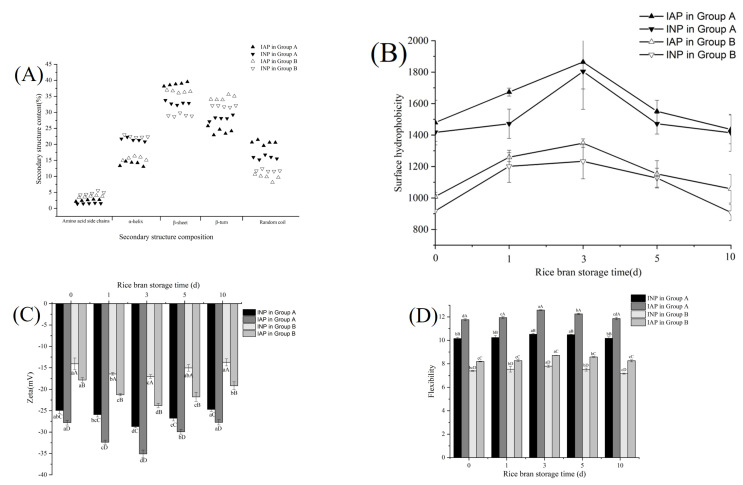
Effects of ultrasonic intensity on the secondary structure (**A**), surface hydrophobicity (**B**), Zeta potential (**C**), and trypsin hydrolysis (**D**) of IAP and INP of RBPE prepared with RB at different storage times, and the different lowercase letters within a column indicate significant differences (*p* < 0.05).

**Figure 4 foods-11-03896-f004:**
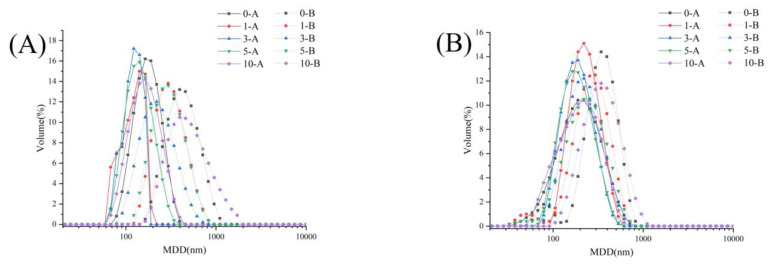
Effects of ultrasonic intensity on particle size distribution ((**A**): particle size distribution of IAP; (**B**): particle size distribution of INP), intrinsic fluorescence spectra ((**C**): intrinsic fluorescence spectrum of IAP; (**D**): intrinsic fluorescence spectrum of INP) and the aggregation morphology ((**E**): non-reductive SDS-PAGE of IAP and INP in group A; (**F**): reductive SDS-PAGE of IAP and INP in group A; (**G**): non-reductive SDS-PAGE of IAP and INP in group B; (**H**): reductive SDS-PAGE of IAP and INP in group B) of IAP and INP of RBPE prepared with RB with different storage times.

**Table 1 foods-11-03896-t001:** Effect of ultrasonic intensity on MDD, PDI, and zeta potential of RBPE prepared by RB with different storage time.

	MDD(nm)					PDI					Zeta(mV)				
	0	1	3	5	10	0	1	3	5	10	0	1	3	5	10
300 W 20 min	1954.33 ± 402.50 ^a^	1378.00 ± 112.79 ^b^	1831.00 ± 59.86 ^a^	505.80 ± 15.56 ^f^	1432.00 ± 17.83 ^a^	0.95 ± 0.06 ^a^	0.89 ± 0.08 ^ab^	0.96 ± 0.07 ^ab^	0.41 ± 0.06 ^c^	0.88 ± 0.03 ^a^	−25.56 ± 1.20 ^a^	−26.83 ± 0.67 ^a^	−26.59 ± 1.49 ^ab^	−29.37 ± 0.32 ^a^	−26.66 ± 1.42 ^a^
300 W 30 min	1594.60 ± 42.69 ^b^	1334.50 ± 55.54 ^bc^	1720.08 ± 122.17 ^ab^	727.53 ± 4.17 ^ef^	1404.00 ± 278.26 ^a^	0.89 ± 0.15 ^ab^	0.80 ± 0.17 ^bcd^	0.98 ± 0.09 ^a^	0.51 ± 0.06 ^c^	0.83 ± 0.11 ^ab^	−26.44 ± 0.94 ^a^	−27.80 ± 0.36 ^ab^	−26.45 ± 1.45 ^ab^	−28.87 ± 0.21 ^a^	−27.17 ± 1.69 ^ab^
300 W 40 min	1584.27 ± 426.38 ^b^	1304.87 ± 309.98 ^bcd^	1724.67 ± 48.21 ^ab^	1153.33 ± 16.04 ^bcd^	1391.85 ± 302.41 ^a^	0.88 ± 0.21 ^abc^	0.77 ± 0.13 ^bcde^	0.96 ± 0.06 ^ab^	0.70 ± 0.00 ^ab^	0.81 ± 0.08 ^ab^	−28.28 ± 2.11 ^b^	−27.70 ± 0.26 ^ab^	−26.43 ± 1.52 ^ab^	−28.47 ± 0.25 ^a^	−27.39 ± 1.06 ^abc^
300 W 50 min	1235.73 ± 126.87 ^c^	1298.00 ± 65.51 ^bcd^	1564.43 ± 195.86 ^bc^	1132.33 ± 20.01 ^bcd^	1350.67 ± 323.03 ^a^	0.85 ± 0.16 ^abc^	0.64 ± 0.19 ^ef^	1.00 ± 0.00 ^a^	0.78 ± 0.09 ^ab^	0.80 ± 0.18 ^ab^	−28.37 ± 1.55 ^b^	−27.82 ± 1.02 ^ab^	−26.43 ± 1.59 ^ab^	−28.30 ± 0.26 ^a^	−27.96 ± 1.41 ^abcd^
400 W 20 min	1187.33 ± 45.01 ^c^	821.73 ± 28.41 ^e^	1785.67 ± 19.55 ^a^	904.20 ± 31.89 ^de^	1336.00 ± 12.53 ^a^	0.86 ± 0.16 ^abc^	0.57 ± 0.08 ^fg^	1.00 ± 0.00 ^a^	0.66 ± 0.04 ^b^	0.80 ± 0.15 ^ab^	−25.67 ± 0.79 ^a^	−27.88 ± 0.96 ^ab^	−25.49 ± 0.59 ^a^	−28.48 ± 1.03 ^a^	−28.22 ± 1.11 ^bcd^
400 W 30 min	1502.60 ± 131.38 ^bc^	603.27 ± 23.03 ^f^	617.23 ± 28.30 ^e^	925.00 ± 48.75 ^de^	967.20 ± 37.70 ^b^	0.86 ± 0.12 ^abc^	0.44 ± 0.02 ^g^	0.59 ± 0.09 ^d^	0.71 ± 0.11 ^ab^	0.80 ± 0.10 ^ab^	−26.48 ± 1.38 ^a^	−28.60 ± 0.61 ^b^	−28.13 ± 0.21 ^bc^	−27.93 ± 0.35 ^a^	−28.46 ± 1.54 ^bcd^
400 W 40 min	1452.63 ± 283.42 ^bc^	739.54 ± 162.23 ^ef^	668.86 ± 102.19 ^e^	947.60 ± 28.28 ^cde^	952.40 ± 31.12 ^b^	0.78 ± 0.12 ^bc^	0.67 ± 0.11 ^def^	0.59 ± 0.08 ^d^	0.72 ± 0.09 ^ab^	0.79 ± 0.04 ^ab^	−28.36 ± 0.87 ^b^	−28.05 ± 1.49 ^ab^	−29.10 ± 0.90 ^c^	−27.73 ± 0.43 ^a^	−28.59 ± 1.73 ^bcd^
400 W 50 min	1217.50 ± 39.07 ^c^	1132.80 ± 77.62 ^d^	1425.78 ± 217.37 ^c^	1228.33 ± 21.03 ^bc^	832.35 ± 57.47 ^b^	0.83 ± 0.11 ^abc^	0.71 ± 0.17 ^def^	0.90 ± 0.10 ^b^	0.79 ± 0.16 ^ab^	0.78 ± 0.10 ^ab^	−28.60 ± 1.42 ^b^	−27.07 ± 0.67 ^ab^	−25.67 ± 1.47 ^a^	−27.72 ± 1.28 ^a^	−28.74 ± 1.28 ^cd^
500 W 20 min	1284.67 ± 59 ^bc^	1153.75 ± 48.49 ^cd^	1601.14 ± 245.24 ^bc^	1056.89 ± 163.13 ^bcd^	826.96 ± 25.50 ^b^	0.81 ± 0.08 ^abc^	0.73 ± 0.05 ^cde^	0.79 ± 0.07 ^c^	0.76 ± 0.15 ^ab^	0.71 ± 0.08 ^bc^	−25.62 ± 1.12 ^a^	−27.84 ± 0.87 ^ab^	−25.56 ± 0.61 ^a^	−27.86 ± 1.67 ^a^	−28.74 ± 1.56 ^cd^
500 W 30 min	1211.43 ± 411.29 ^c^	1408.33 ± 46.01 ^ab^	687.84 ± 52.45 ^e^	1123.29 ± 45.52 ^bcd^	752.90 ± 21.70 ^b^	0.74 ± 0.17 ^c^	0.78 ± 0.08 ^bcde^	0.62 ± 0.08 ^d^	0.80 ± 0.05 ^ab^	0.68 ± 0.08 ^bc^	−26.53 ± 1.03 ^a^	−27.70 ± 0.73 ^ab^	−27.00 ± 1.15 ^ab^	−27.56 ± 1.51 ^a^	−28.84 ± 1.34 ^cd^
500 W 40 min	853.88 ± 132.50 ^d^	1459.47 ± 102.79 ^ab^	641.66 ± 19.19 ^e^	1321.00 ± 82.40 ^b^	948.80 ± 30.25 ^b^	0.54 ± 0.09 ^d^	0.88 ± 0.1 ^abc^	0.57 ± 0.1 ^d^	0.83 ± 0.17 ^ab^	0.63 ± 0.04 ^cd^	−29.22 ± 1.58 ^b^	−27.51 ± 0.77 ^ab^	−27.50 ± 0.20 ^bc^	−27.59 ± 1.26 ^a^	−29.11 ± 0.90 ^d^
500 W 50 min	675.25 ± 26.07 ^d^	1578.43 ± 85.77 ^a^	943.17 ± 176.54 ^d^	1679.27 ± 343.73 ^a^	682.57 ± 16.55 ^b^	0.51 ± 0.03 ^d^	0.96 ± 0.01 ^a^	0.82 ± 0.09 ^c^	0.85 ± 0.20 ^a^	0.52 ± 0.02 ^d^	−29.31 ± 1.00 ^b^	−27.04 ± 1.64 ^ab^	−27.14 ± 1.13 ^ab^	−27.40 ± 0.80 ^a^	−29.35 ± 0.33 ^d^

The different lowercase letters within a column indicate significant differences (*p* < 0.05).

**Table 2 foods-11-03896-t002:** Effects of ultrasonic intensity on the content of IAP and free sulfhydryl and disulfide bonds of IAP and INP.

	Γ (mg/m^2^)		Free Sulfhydryl (nmol/mg)				Disulfide Bonds (nmol/mg)			
	Group A	Group B	IAP in Group A	INP in Group A	IAP in Group B	INP in Group B	IAP in Group A	INP in Group A	IAP in Group B	INP in Group B
0	11.95 ± 0.00 ^b^	24.45 ± 0.28 ^b^	39.91 ± 0.48 ^c^	34.70 ± 0.65 ^a^	24.14 ± 0.41 ^c^	20.59 ± 0.36 ^a^	32.38 ± 1.78 ^a^	20.95 ± 0.29 ^c^	19.22 ± 0.97 ^ab^	9.80 ± 0.36 ^e^
1	11.67 ± 0.01 ^c^	24.04 ± 0.68 ^b^	41.94 ± 0.69 ^b^	33.42 ± 0.50 ^a^	25.38 ± 0.54 ^b^	18.81 ± 0.67 ^b^	25.57 ± 5.04 ^b^	22.71 ± 0.11 ^b^	17.78 ± 0.48 ^c^	11.02 ± 0.63 ^d^
3	11.10 ± 0.02 ^c^	23.18 ± 0.12 ^c^	44.93 ± 1.07 ^a^	30.19 ± 0.91 ^b^	27.33 ± 0.46 ^a^	16.30 ± 0.62 ^c^	28.98 ± 1.17 ^ab^	23.30 ± 0.37 ^b^	16.54 ± 0.17 ^d^	13.25 ± 0.08 ^c^
5	10.88 ± 0.01 ^d^	22.49 ± 0.09 ^d^	41.56 ± 0.94 ^b^	29.34 ± 0.72 ^b^	25.77 ± 0.25 ^b^	13.90 ± 0.39 ^d^	30.30 ± 0.35 ^a^	24.59 ± 0.35 ^a^	18.53 ± 0.16 ^bc^	16.24 ± 0.47 ^b^
10	11.98 ± 0.01 ^a^	26.26 ± 0.15 ^a^	40.01 ± 0.87 ^c^	27.77 ± 0.71 ^c^	24.29 ± 0.49 ^c^	11.30 ± 0.34 ^e^	32.11 ± 0.25 ^a^	25.57 ± 1.07 ^a^	19.90 ± 0.43 ^a^	17.88 ± 0.11 ^a^

The different lowercase letters within a column indicate significant differences (*p* < 0.05).

**Table 3 foods-11-03896-t003:** Correlation of the stability of RBPE prepared from RB with different rancidity extents and the structural characteristic of IAP and INP.

		MDD	PDI	RBPE Zeta-Potential	CI
IAP	α-helix/β-sheet	−0.87 **	−0.85 **	−0.78 **	−0.86 **
	Zeta-potential	0.87 **	0.84 **	0.77 **	0.79 **
	Disulfide bonds	−0.69 *	−0.69 *	−0.83 **	−0.68 **
	Flexible	−0.94 **	−0.95 **	−0.89 **	−0.87 **
	Surface hydrophobicity	−0.82 **	−0.77 **	−0.70 **	−0.79 *
	Particle size distribution	0.83 **	0.83 **	0.84 **	0.78 **
	λmax	−0.42	−0.41	−0.45	−0.57
INP	α-helix/β-sheet	−0.93 **	−0.94 **	−0.92 **	−0.89 **
	Zeta-potential	0.93 **	0.92 **	0.86 *	0.89 **
	Disulfide bonds	−0.84 **	−0.88 **	−0.77 **	−0.95 **
	Flexible	−0.94 **	−0.95 **	−0.89 **	−0.89 **
	Surface hydrophobicity	−0.82 *	−0.76 *	−0.73*	−0.81 **
	Particle size distribution	0.65 *	0.70 *	0.77*	0.54
	λmax	−0.63	−0.60	−0.58	−0.54

* *p* < 0.05, ** *p* < 0.01.

## Data Availability

All available data are contained within the article.

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
