# Peer review of "Strategy and Mechanism of Rice Bran Protein Emulsion Stability Based on Rancidity-Induced Protein Oxidation: An Ultrasonic Case Study"

_foods, 2022, doi:10.3390/foods11233896_

Round 1

Reviewer 1 Report

Dear authors,

this is a interesting study. The background of this study and methodology are well written, and good explained. The manuscript has a lot of tehnical errors (like different fonts in titles and subtitiles; some subtitles are in italic some not; different fonts in Figueres titles; line 191 letter "B"???; figueres (Fig 4.)-overlap letters). The manuscript needs improvement. 

- in the abstract- first sentence (line 11-12)-not clear. Is that the aim of tne study??

Reviewer 2 Report

This manuscript entitled “Strategy and mechanism of rice bran protein emulsion stability based on rancidity-induced protein oxidation: an ultrasonic case studyis interesting from the scientific and market point of view. The given information regarding rice bran protein (RBP) and prepared emulsions in this manuscript is useful for researchers and academia and article would be great contribution in discipline of food science and food chemistry.  The room for improvement is always there and I have suggested some minor revisions. I can extend my services to further review the incorporation of the corrections in article again.

·         Abstract-mention numerical value for better understanding of the reader.

·         P-1 (L-12-13) Recheck this statement- The ultrasonic conditions were optimized according to the results of the mean droplet diameter, polydispersity index, zeta potential, and creaming index of the RBPE prepared from oxidized RBP..

·         L-41 Cross heck the reference? The moderate oxidation of RBP could cleave the disulfide bonds, enhance the flexibil- 41 ity of RBP structure, and promote the unfolding of RBP structure

·         Variety selection criteria?

·         L-76 Any criteria for treatment - Fresh RB was stored for 0, 1, 3, 5, and 10 d to prepare the RB with different rancidity 76 extents (25℃, relative humidity 85%)

·         -19 Addition-ally, while mean intakes of red and products, poultry and game were above recommended levels, mean intakes of fruits, vegetables, fish, and shrimp were below recommended levels)

·         L-298-299 Need clarity- The α-helix content of group A was sig- 298 nificantly lower than that of group B, but the β-sheet content of group A was significantly 299 higher than that of group B (P<0.05).

·         L-340-Any relevancy The phenomenon might be due to the fact that highly unfolded pro- tein was more prone to adsorb to the oil-aqueous interface

·         L-408- Creating no sense???????????? The MDD, PDI, zeta potential, and CI of RBPE were significantly posi-  tively correlated with zeta potential, and particle size distribution of IAP and INP (P < 409 0.01), and negatively correlated with the α-helix/β-sheet ratio, disulfide bonds content 410 flexibility, and surface hydrophobicity, of IAP and INP (P < 0.01)?.

·         Cross check methods & cited references in  3..2.1 , 3.2.2 3.2.4. and3. 2.5

·         Cite the following latest papers

o    Wang, Weining, et al. "Effect of ultrasonic power on the emulsion stability of rice bran protein-chlorogenic acid emulsion." Ultrasonics Sonochemistry 84 (2022): 105959.

o    Wang, S., Wang, T., Li, X., Cui, Y., Sun, Y., Yu, G., & Cheng, J. (2022). Fabrication of emulsions prepared by rice bran protein hydrolysate and ferulic acid covalent conjugate: Focus on ultrasonic emulsification. Ultrasonics Sonochemistry, 88, 106064.

o    Liu, J., Gao, T., Li, F., & Xie, T. (2022). The addition of oxidized tea polyphenols enhances the physical and oxidative stability of rice bran protein hydrolysate-stabilized oil-in-water emulsions. Food Science and Technology Research, 28(3), 225-233.

·         Please avoid repetition-

·         Please check reference style throughout MS

·         Italic all the scientific names,

·         Remove grammatical mistakes

·         Need to rewrite the conclusion

·         Recheck Legends description is as per figure number and discussion-

·         I urge the authors to improve the English language for better flow of literature.

Reviewer 3 Report

Line 11 & 12: The verb of the sentence? Changed to “To improve the practical utilization of rice bran protein (RBP), it was extracted from rice bran (RB) with different rancidity extents as an emulsion stabilizer.”
Materials & methods: The space between the numerical value and unit of temperature should be considered (For example 25 °C)
Line 82: stringing change to “stirring”
Figures: The figures were not clear enough, this may be due to a submission issue, please enlarge all of the axis titles and numbers and improve the quality of the figures. Moreover, the figures' names (i.e., A, B, C, etc) were very far from the corresponding figures in the submitted manuscript, Please check all of them.
Line 276: comma after Table 2
Line 295, 314, and 383: Fig. 3A (or Fig. 3B, …) shows
